# The Fifth Dimension in Socio-Scientific Reasoning: Promoting Decision-Making about Socio-Scientific Issues in a Community

Hava Ben-Horin [1,*], Yael Kali [1] and Tali Tal [2,3]

1 Department of Learning and Instructional Sciences, University of Haifa, 199 Aba Khoushy Ave. Mount Carmel, Haifa 3498838, Israel
2 Faculty of Education in Science and Technology, Technion—Israel Institute of Technology, Haifa 3200003, Israel
3 Samuel Neaman Institute for National Policy, Haifa 3200003, Israel
* Correspondence: hava.abramsky@gmail.com

**Abstract:** Making informed decisions about socio-scientific issues requires socio-scientific reasoning, which is highly challenging for students. This construct has four dimensions, including recognizing complexity, analyzing multiple perspectives, appreciating the need for ongoing inquiry, and employing skepticism. To support its development, we integrated established principles for designing socio-scientific issues learning environments with the Knowledge Community of Inquiry model. A design-based research study was conducted with two iterations, involving 85 eighth graders, to examine the effectiveness of the integrative approach in supporting students' socio-scientific reasoning and decision-making. The web-based unit "Asthma in the Community" was adapted and enacted with the students. In the first iteration, the socio-scientific approach was applied, and in the second iteration, the unit was redesigned with features from the Knowledge Community of Inquiry approach. Results showed that all students who participated in the second iteration developed socio-scientific reasoning, made better evidence-based decisions than those in the first iteration, and expressed an understanding of decision-making processes in a community, which is suggested as new, fifth dimension of socio-scientific reasoning. This fifth dimension is suggested as essential for coping with decision-making in socio-scientific issues in the networked society, and the study suggests how to design learning environments that can support its development.

**Keywords:** decision-making; socio-scientific issues (SSIs); socio-scientific reasoning (SSR); science education; Knowledge Community of Inquiry (KCI); design-based research (DBR)

## 1. Introduction

Life in the 21st century's networked society requires citizens to make decisions about socio-scientific issues (SSIs) that may influence personal, family, community, and sometimes global matters [1–3]. Many socio-scientific issues are related to sustainability. Some examples are climate change, biodiversity loss, changes in urban environments, resource allocation, and poverty. These issues significantly impact both present and future generations, as well as economic systems [4], citizen engagement, and political systems, which identify them as socio-scientific. All environmental debates and public health concerns are inherently related to sustainability because of their multi-faceted, complex, and controversial nature. Many people rely on the Internet and social networks, sometimes as a main source of information, to make important decisions [2,5]. The Internet has provided unparalleled access to vast amounts of information and has enabled individuals to create and share information with others. However, it can also be considered a "double-edged sword", presenting new challenges such as the need to evaluate multiple information sources that may be contradictory, biased, or misleading (fake news) and dealing with information bubbles that tend to limit the exposure to various perspectives on matters of interest [1,2,6]. Thus, it is hard for citizens with only basic scientific education to make

informed decisions, resulting with reliance on "gut feelings" or adoption of misinformation and sometimes disinformation without applying critical thinking [1,2]. Consequently, there is an urgent need to educate young people to meaningfully participate in discourse and decision-making concerning socio-scientific issues both face-to-face and on the Internet and social networks as responsible citizens. In the context of socio-scientific issues—the focus of this special issue—well-designed learning environments play a crucial role in providing students with the knowledge, skills, and mindsets necessary to navigate such complexity and make informed decisions that consider both scientific and societal perspectives [7]. Such environments will need to include scaffolds and tools for students to evaluate arguments in the information they consume [6,8].

The SSI approach, in which students learn science in the context of socio-scientific issues [6,9], provides a means to develop students' ability to participate in democratic decision-making on issues that involve conflicts for citizens and have scientific, technological, and environmental aspects that are relevant to their lives. This approach enhances education for sustainability by fostering a deep understanding of the interconnectedness between human actions and the environment, empowering learners to contribute to a more sustainable future [10]. To address SSIs effectively, students must apply relevant scientific knowledge and a range of higher-order thinking skills [11], including the ability to base their decisions on evidence and socio-scientific reasoning (SSR). SSR involves several aspects that require epistemic knowledge of the characteristics of SSIs. This includes an understanding of the complexity of these issues and their ongoing inquiry, the ability to consider them from multiple perspectives, and the capacity to approach information from various stakeholders on these issues with skepticism [12].

Previous studies have demonstrated the potential of the SSI approach to enhance students' scientific knowledge [13–15] and argumentation skills [15–17]. However, the literature also indicates that developing students' socio-scientific reasoning can be challenging [9,14] and that students may struggle to support their decisions with evidence [12,15,17]. While the potential of collaborative frameworks for argumentation and decision-making has been mentioned in the literature [17,18], there is limited research on decision-making related to socio-scientific issues within communities.

In this study, we push forward the field's understanding of SSR and decision-making concerning SSIs by introducing new strategies that integrate relevant innovative pedagogical approaches to tackle the unique challenges of decision-making in a networked society.

We adopted the pedagogical model of Knowledge Community of Inquiry (KCI) [19], which combines the approaches of learning communities with inquiry-based learning. In this model, students in a classroom function as a community, which collectively conducts continuous, guided inquiry of a complex phenomenon.

In this study, we integrated the KCI and SSI approaches to develop a synthesized approach (KCI-SSI) that involves students collaborating and making decisions on socio-scientific issues while acting as a knowledge community. We redesigned an existing Web-based curricular unit "Asthma in the Community" [20], using the KCI-SSI approach, and implemented it through two iterations of design-based research with 85 eighth graders. In iteration 1, we applied the SSI approach, while in the second, we added KCI features.

The aims of our study were to understand how students' socio-scientific reasoning and decision-making abilities develop and to assess the contribution of the KCI-SSI approach to this process. Our research questions were:

1. Research Question 1 (RQ1). What are the characteristics of socio-scientific reasoning that students develop in a KCI-SSI designed learning environment?
2. Research Question 2 (RQ2). How does the KCI-SSI approach contribute to students' decision making about socio-scientific issues?

## 2. Theoretical Background

### 2.1. Scientific Literacy and the Aim of Science Education

The term "scientific literacy" is typically used to answer the question: "What should the public know about science?" [21]. The term has many interpretations that arise from different views on the nature of the necessary knowledge and the definition of "the public" [22]. Roberts [23] identified two opposing views on scientific literacy. The first view, Vision I, argues that science education should focus on scientific content and that scientifically literate individuals should have knowledge and practices derived solely from scientific fields, without reference to non-scientific contexts. The second view, Vision II, emphasizes learning scientific content in the context of everyday situations where science plays a key role. According to this vision, scientifically literate individuals will be able to apply scientific knowledge and practices in socio-scientific contexts in their everyday lives, requiring a deep understanding of science and the development of transfer skills [24,25].

Contemporary views of scientific literacy align with Vision II, as seen in the OECD 2015 framework, which defines scientific literacy as "the ability to engage with science-related issues, and with the ideas of science, as a reflective citizen" [26], and in the K-12 Framework for Science Education in the United States [27], which expects students to have sufficient knowledge of science and engineering to engage in public discussions on science-related issues and to be critical consumers of scientific information related to their everyday lives by the end of 12th grade. In this study, we adopted Roberts' Vision II and contemporary perspectives on scientific literacy.

### 2.2. The Socio-Scientific Approach to Science Education

The socio-scientific approach offers an alternative for traditional science education—preparing students for their role as future citizens, rather than as future scientists [23]. Learning in the SSI approach involves students in processes of inquiry, discourse, and decision-making on socio-scientific issues [28,29]. The SSI approach invites students to engage in discourse and knowledge-building in the context of real-world issues, thereby enhancing their ability to participate in democratic decision-making processes [30]. The SSI approach offers a genuine context in which interdisciplinary and often controversial issues can be studied, recognizing that many problems cannot be solved merely by attending to scientific evidence alone [31]. In a special issue of *Research in Science Education*, focused on SSIs and sustainability in science education, the editors highlight the ways in which SSI pedagogy serves the interests of both education for sustainability and science education. They argue that science education must change to tackle the social, economic, cultural, and environmental challenges we face [32]. Additionally, the editors claim that "while exploring the intersections of science, society, environment and sustainability within educational contexts, the large-scale tension being explored is that between commitment to the interests of disciplinary science and commitment to the interests of sustainable communities". A decade later, Van Der Leij et al. [33] asked how schools and teachers can meaningfully engage their students in challenges reflected by the United Nations' 17 Sustainable Development Goals (SDGs) and explored various methods for using morality, values, ethics, and socio-scientific issues. They concluded that SSI-based teaching is an effective means of addressing wicked sustainability issues within science education. An example of how this can be achieved is by leveraging the SSI approach's ability to facilitate discussions on multiple solutions to complex environmental issues while allowing individuals to refine and justify their own perspectives on these issues. This process may ultimately lead to the development of responsible scientific literacy and citizenship [34]. Given the contemporary and future global challenges, such as the COVID-19 pandemic and global warming, as well as the growing need to thoughtfully navigate post-truth challenges [6,8], the SSI pedagogical approach gains even greater significance.



### 2.3. Socio-Scientific Reasoning (SSR)

What do students need to successfully tackle and resolve socio-scientific issues? Inquiry and decision-making in complex socio-scientific issues require students to apply relevant scientific knowledge and understanding of the nature of socio-scientific issues [35,36]. Additionally, they need a wide range of higher order thinking skills such as the ability to plan and conduct investigations, obtain and interpret new information and data, evaluate evidence from different sources, participate in argumentation processes, propose possible solutions, identify risks and uncertainties, and compare and weigh the advantages and disadvantages of various solutions while considering various factors such as environmental, social, and moral implications [35,36]. Many of these skills align with the scientific practices outlined in the NRC documents [27], such as planning and conducting investigations, analyzing and interpreting data, engaging in argument from evidence, and obtaining, evaluating, and communicating information.

Sadler and his colleagues [12] coined the unique concept of socio-scientific reasoning (SSR), which relates to the inherent features of socio-scientific issues and describes a set of practices needed for their resolution. These practices include: (1) recognizing the inherent complexity in socio-scientific issues, (2) analyzing them from multiple perspectives, (3) appreciating the need for their ongoing inquiry, and (4) employing skepticism of potentially biased information. Socio-scientific reasoning is anchored to relevant scientific knowledge and refers to some of the higher-order competencies listed before and to the knowledge about the nature of socio-scientific issues. The SSR construct corresponds with Lederman and his colleagues [37], who claim that in order for a person to make informed decisions regarding issues related to science, they must consider the claims and evidence in light of the characteristics inherent in the nature of scientific knowledge and scientific research. In this research, we have chosen to adopt the SSR construct as one of the desired learning outcomes within the SSI approach.

### 2.4. Effect of SSI Instruction on Argumentation, Decision-Making, and SSR

A growing body of research has emerged in the last decade related to the contribution of learning in the SSI approach to several aspects of scientific literacy. This includes scientific knowledge, argumentation practices, logical thinking, knowledge about science, positive attitudes towards science, and more [7,12].

Regarding argumentation, the SSI approach holds potential for enhancing learners' argumentation skills by facilitating their participation in discourse on complex issues and exposure to diverse positions [12,17]. Though the SSI approach can promote argumentation, its success is dependent on supplying appropriate scaffolds to learners [15,16,35].

Decision-making in SSIs poses a cognitive challenge to learners, which can "get lost" due to the complexity of the issue itself, along with the complexity of the decision-making process [38]. These challenges can be relieved using appropriate scaffolds for the decision-making process and collaborative learning and interweaving the process of learning scientific knowledge together with the decision-making process [20,38].

The literature dealing with the effect of SSI instruction on SSR development is underdeveloped and highlights difficulties both in the measurement of SSR and in its achievement [14]. SSR is usually measured using a questionnaire that enables quantitative measurement of the different practices of socio-scientific reasoning (quantitative assessment of SSR—QaSSR). The questionnaire includes socio-scientific scenarios different from those studied by the students, followed by a series of open questions [14]. Results show no significant contribution to SSR in short-term SSI interventions [9,14], with some success in longer-term interventions [39,40].

The difficulties that learners face in decision-making within SSIs can be attributed to the challenge of acquiring epistemic knowledge about the nature of science [41], which is central to the SSR construct. Research has shown that the active participation of learners in scientific inquiry, along with explicit and reflective instruction on the nature of science, can support the development of this knowledge [41,42]. Similarly, Sadler and his colleagues [9]

emphasize the significance of exposing learners to different SSIs, while explicitly referring to their shared characteristics.

Furthermore, the literature addresses the challenges that students face in coping with others' opinions and the tendency to adhere to their own opinions [43,44], which may contribute to the difficulties in developing the multiple perspectives aspect of SSR. This might be connected to how well students can adapt to different ways of understanding the world, which is known as "epistemic fluency" [45]. This skill is very important for discussing real-world problems in the 21st century [46]. The research literature regarding discussions in a democracy has shown that students who possess more complex epistemic conceptions and exhibit critical and open-minded thinking are more likely to change their minds and adopt new ideas [47]. Moreover, this literature highlights the importance of discussions on controversial issues where participants have "reasonable disagreements" [48]. In such discussions, individuals provide reasons for their viewpoints and show respect for others' perspectives. Necessary conditions for these discussions, include assigning of roles and speaking rights based on turn-taking, nurturing of norms where everyone can express their voices, and are open to negotiation and influence from well-supported positions. Thus, a pedagogical approach that supports collaboration seems to have the capacity to effectively address the challenges of SSR development, promote epistemic fluency, and encourage the formation of "reasonable disagreements" among students.

While there are several models for SSI instruction described in the literature [49–51], there is a lack of coherent and clear design knowledge [7], based on interventional research, that can guide educators in designing effective SSI-based learning environments.

*2.5. The Knowledge Community of Inquiry (KCI) Model*

As described before, one way to support students in the challenging processes of inquiry, argumentation and decision-making concerning complex socio-scientific issues, is to adopt collaborative learning scripts, which are a characteristic of learning communities. In learning communities, the whole class acts as a community, and inquiry and knowledge building are carried out as a community endeavor, rather than as an individual or a small group effort [52]. An important aspect of learning in a community is the accountability that it can foster among students and the gradual development of accountable talk [53]. The latter is characterized by responsibility that participants develop towards the community, towards norms of justification, and towards knowledge [53]. A discussion of interesting, relevant, and complex socio-scientific issues, which provide a variety of perspectives and solutions, can serve as fertile ground for the development of an accountable discourse, with appropriate supports. Moreover, the accountable discourse itself may contribute to the advancement of knowledge, evidence-based decision-making, and consideration of a variety of perspectives on socio-scientific issues. The Knowledge Community and Inquiry (KCI) pedagogical model integrates structured inquiry with ideas from different models of learning communities, such as enabling students to collectively build their knowledge, giving voice and shared authority to students, and distributing expertise. In this model, the whole class collectively conduct continuous, guided, and scaffolded inquiry of a complex phenomenon, directly related to the learning goals [19,54]. The collective inquiry artifacts accumulate in a collective knowledge base that is accessible to all community members for further inquiry and is continuously negotiated and revised by them. The knowledge base is situated in a technology-mediated environment that scaffolds students as they add ideas and revise or synthesize them into new ideas and arguments [19]. In the KCI model, students study complex, ill-structured phenomena, which provide many possibilities for inquiry, expression of multiple perspectives, and integration of scientific knowledge [54–56].

We assume that integrating the SSI and KCI approaches can promote students' SSR and their ability to make evidence-based decisions. This assumption is based on (a) the inherent potential of KCI to support and scaffold ongoing collaborative inquiry of complex phenomena, (b) the potential of collaborative processes to support argumentation [17,38],

(c) the accountable talk that is supported within a learning community [52,53], and (d) the potential of the KCI model to support the expression of multiple perspectives by all community members [56].

## 3. Design of the Learning Unit

The original "Asthma in my community" unit, as well as the adaptations made in the current study, were developed using the Web-based Inquiry Science Environment (WISE) platform [19]. The students in WISE are guided in the inquiry process, individually or in small groups.

### 3.1. Iterations and Participants

The original unit was redesigned in two iterations (Table 1) with the participation of eighth graders (aged 13–14) from a leading private school in Israel that focuses on science and environmental studies. The students were guided by their science and geography teachers during the learning unit. The group of students is not representative of eighth graders in Israel in terms of socio-economic status, but the classes were diverse in terms of achievements and learning disabilities.

**Table 1.** Iterations and participants.

| Iteration | Participants | Design Revisions to Original WISE Unit |
|:---:|:---:|:---:|
| 1 | 46 eighth graders | employing SSI approach |
| 2 | 39 eighth graders | employing KCI-SSI approach |

### 3.2. Special Features Used

In WISE, one tool we identified as having the potential for supporting collective knowledge-building and decision-making is the "idea basket". This feature allows students to add, share, and collaboratively refine ideas encountered during learning a WISE unit. Ideas collected in the basket can later be used by students to construct arguments. In the study, we utilized this feature as an evidence basket for both within-team learning (iterations 1 and 2) and between-team community learning (iteration 2). In the latter case, the evidence basket served as part of the community knowledge base. At different stages of the learning process, students were asked to select evidence from their team evidence basket and share it in the public (community) evidence basket (Figure 1). Students could also choose appropriate evidence from the community evidence basket and incorporate it into their own evidence basket.

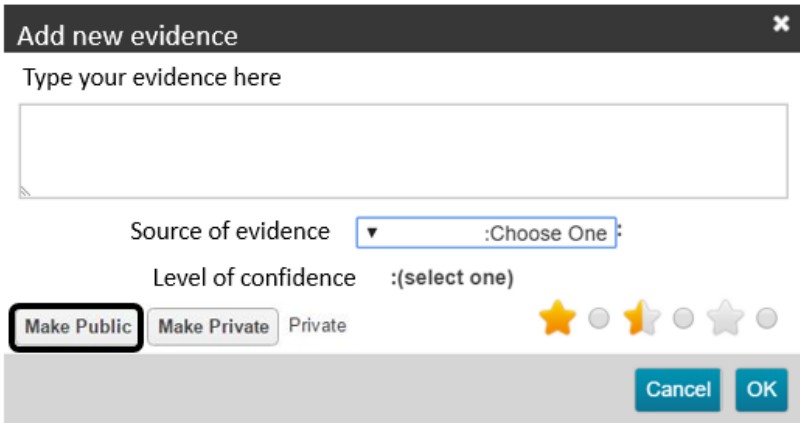

**Figure 1.** Add evidence to evidence basket—private (team) and public (community) options.

A non-WISE feature that we embedded into WISE to augment the knowledge base in iteration 2 was a collective Google Map, where the evidence about the different environmental factors affecting asthma, which the students collected during the collaborative inquiry at different regions, gradually accumulated (Figure 2).

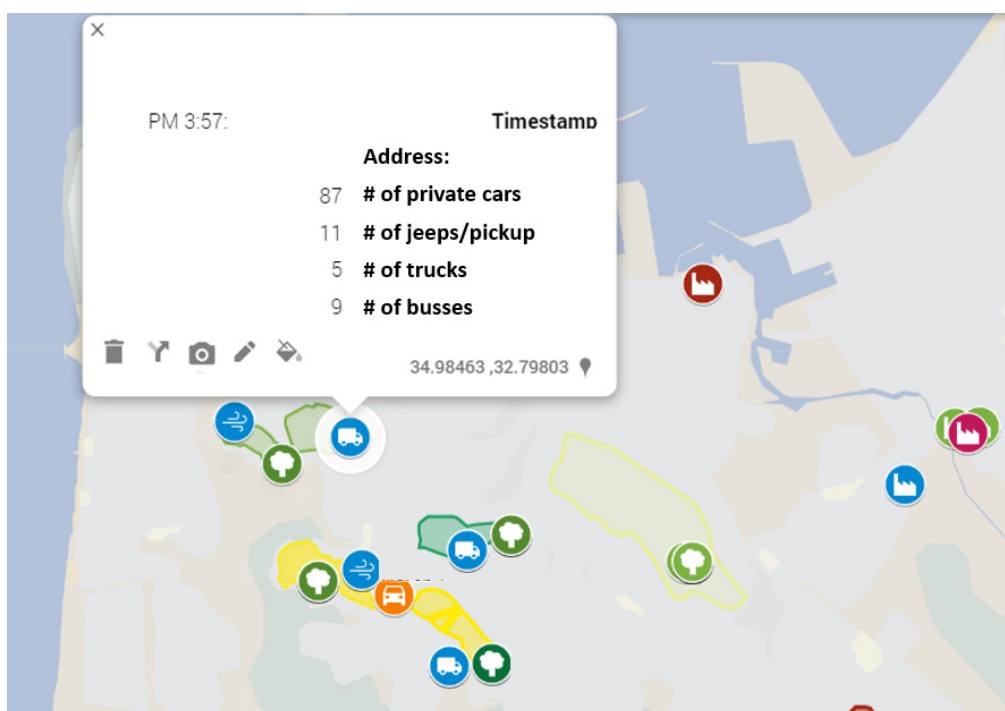

**Figure 2.** Collective map.

*3.3. Sequence of Activities in the Two Iterations*

The learning sequence in both iterations was similar and included students' preliminary acquaintance with the socio-scientific issue of asthma and the environmental factors affecting it, followed by a set of inquiry and decision-making activities and a culminating activity. In iteration 1, we applied the SSI approach, and students conducted the inquiry and decision-making activities in couples, gathering their evidence in a team evidence-basket. In iteration 2, we applied the KCI-SSI approach, adding KCI design elements to the unit. The main design element that was introduced into to the unit in iteration 2 was a jigsaw activity [57] in two phases. The first phase included a collaborative inquiry script where students worked in groups of four as "regional experts". As such, they collected data about different factors affecting asthma in their neighborhood and gathered it in the collective knowledge base—the evidence basket and the collective Google Map. The second phase was a scripted decision-making process in mixed groups of eight students representing different regions. We illustrate the differences between the two iterations in Figure 3.

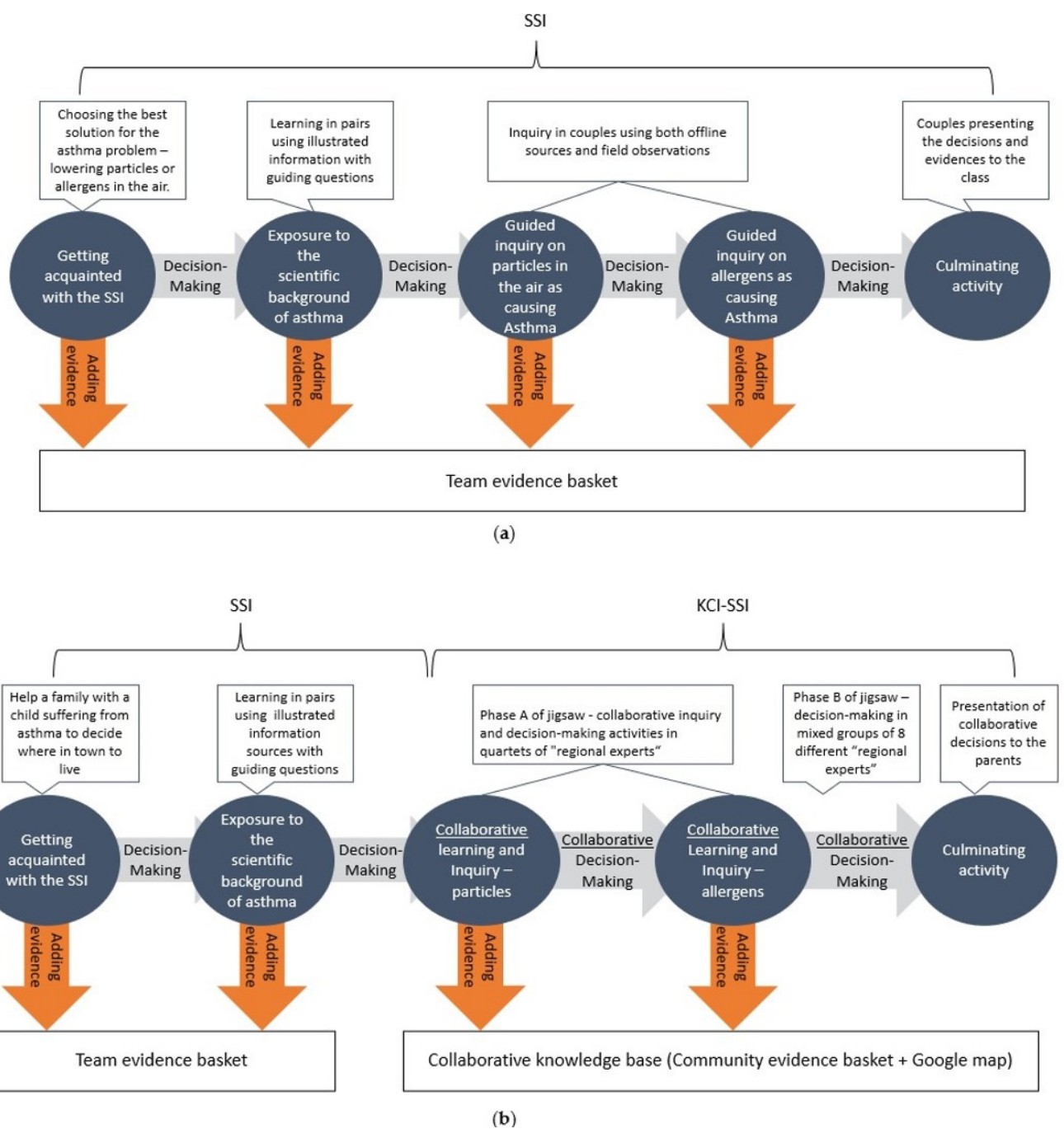

**Figure 3.** Sequence of activities in iterations 1 and 2. (**a**) sequence of activities in iteration 1, applying SSI features; (**b**) sequence of activities in iteration 2, additionally applying KCI and KCI-SSI features.

## 4. Methodology

### 4.1. Methodological Approach

This study was conducted using a design-based research (DBR) approach, which emphasizes the complex context in which learning takes place and can examine the effect of curricular innovations on learning in its natural context [58,59]. DBR has two main interrelated goals: (1) theoretical contribution—understanding of learning processes by examining the impact of design elements in learning environments on specific learning processes and (2) practical contribution—insights into how learning can be promoted through innovative learning environments [7]. Design research is interventional research, which is carried out in several rounds (iterations), each of which includes the design of

the learning environment, its enactment in its natural context (such as a classroom), the collection and analysis of data, and the refinement of the design following the findings for the next round [7,58,59]. Over time, design-based research has gained recognition as a legitimate form of research in the fields of education and learning sciences, leading to the creation of various learning environments, many of which incorporate technology to enhance the learning experience [7,58]. Using DBR enabled us to address our dual goal of advancing both learning theory regarding SSR (RQ1) and design knowledge regarding how to design learning environments that can support student decision-making about SSIs (RQ2).

### 4.2. Data Sources and Analysis

In the study, we collected rich data through: (a) semi-structured, in-depth interviews with six students (only in iteration 2) and (b) written digital artifacts produced by students in WISE (referred to as WISE data). We employed a mixed-methods approach to analyze the collected data, combining qualitative and quantitative methods. Table 2 presents an overview of the data sources and means of analysis, aligned with the research questions in both iterations.

**Table 2.** Data sources and means of analysis for each research question.

| Research Question | Data Sources | Data Analysis |
|---|---|---|
| RQ1. What are the characteristics of socio-scientific reasoning that students develop in a KCI-SSI designed learning environment? | Post semi-structured, in-depth interviews with six students at the end of iteration 2 | Inductive analysis of the interviews followed by deductive analysis according to the SSR framework of Sadler et al. [12] Quantification of the qualitative data [60] |
| RQ2. How does the KCI-SSI approach contribute to students' decision making about socio-scientific issues? | WISE Data: (1) Evidence collected by students in the team evidence basket (2) Evidence collected by students in the community evidence basket (Iteration 2) (3) Source of evidence (4) Decisions (and changes in decisions) regarding the issue | Quantitative analysis of the number of pieces of evidence in the evidence-basket, their distribution and number of changes in decisions Qualitative analysis of the quality of evidence using McNeill and Krajcik [61] framework Comparison between the quality of the first and last decision in each iteration Non-parametric statistical analysis of the data (Wilcoxon test) (only in iteration 2) |
| | Post semi-structured, in-depth interviews with six students at the end of iteration 2 | Inductive analysis of the interviews followed by deductive analysis according to the design elements in the SSI, KCI, and KCI-SSI approaches Quantification of the qualitative data [60] |

### 4.2.1. Semi-Structured Interviews

As presented in Table 2, at the conclusion of iteration 2, we carried out semi-structured, in-depth interviews with six students. This was performed to answer the first research question and to reinforce our answer to the second research question. We chose to employ the in-depth interview tool to assess the socio-scientific reasoning of students in iteration 2, following a pilot study in iterations 1 and 2 using the previously described QaSSR tool, which did not yield meaningful findings. The in-depth interviews allowed us to obtain detailed data and gain a comprehensive understanding of the students' socio-scientific reasoning, which would have been challenging to achieve with questionnaires alone.

The interviews lasted approximately 20 min and involved both male and female students with varying academic accomplishments, degrees of involvement in the project, and contributions to collaborative efforts. During the interviews, we posed reflective questions to the students regarding their experiences in the project. These questions included eliciting key insights, identifying activities that were most influential in their learning, and exploring how they might tackle comparable issues in the future. To assess students' transfer skills concerning SSR, we also asked them to draw parallels between

different socio-scientific issues (SSIs) raised during the interview and to provide examples of similar issues.

The analysis of the interviews employed a qualitative interpretive approach, combining both inductive and deductive methods. Initially, an inductive approach was used to identify themes that emerged from the interview data. Themes were defined as ideas that appeared in the responses of two or more students.

Subsequently, the identified themes were analyzed deductively by applying a socio-scientific reasoning (SSR) framework [12] to address Research Question 1. Additionally, the design elements in the SSI, KCI, and KCI-SSI approaches were utilized to answer Research Question 2. A socio-scientific reasoning (SSR) framework [12] was used to identify themes belonging to the four dimensions of SSR: *complexity*, *ongoing inquiry*, *multiple perspectives*, and *skepticism*. The analysis process entailed identifying literature-based characteristics associated with each dimension of the SSR framework.

1. **Complexity.** Socio-scientific issues are difficult to resolve, as they do not have a clear single solution, are influenced by multiple factors, and have a wide range of impact. They require a choice between options and a cost–benefit analysis, considering the pros and cons of each solution [12,36].
2. **Inquiry.** Socio-scientific issues are problems that do not have a clear solution; they are subject to ongoing inquiry (science in the making), and, as a result, available information may be partial and incomplete. Further inquiry may reveal new information that is necessary for making decisions [12]. A high level of the inquiry dimension involves understanding the uncertainties surrounding socio-scientific issues, recognizing the need for information to address these issues, and being able to identify potential research directions or the information needed to make a decision on these issues.
3. **Multiple Perspectives.** Different stakeholders have different positions on socio-scientific issues based on their personal priorities, professional backgrounds, values, and preconceptions [12]. A high level of the perspectives dimension involves the ability to distinguish and detail the different perspectives of various stakeholders on a socio-scientific issue.
4. **Skepticism.** A high level of skepticism involves referencing the source of information and expressing skepticism towards potentially biased information. A low level of skepticism involves either not referencing the source of information or simply accepting the information without questioning its reliability [12,62]. We treated the need for information from experts as a reference to the source of the information and its reliability.

In order to address Research Question 2, we examined the themes and identified references to different design elements that supported student learning. These references were categorized based on the three approaches utilized in designing the unit: (a) the socio-scientific approach (SSI), (b) the Knowledge Community of Inquiry (KCI) approach, and (c) the integrative KCI-SSI approach. The analysis was conducted in two rounds, initially by the first author, followed by the co-authors. Any disagreements were discussed until consensus was achieved, with a minimum threshold of 80% agreement. After refining the identified themes, a second round of analysis was performed, resulting in 90% agreement among the authors. The first author completed the analysis of the agreed-upon data.

### 4.2.2. WISE Data

To answer the second research question, we relied primarily on data sources from the WISE platform, which included students' responses to open-ended questions, arguments, decisions, and pieces of evidence saved or shared with others through the evidence basket. We analyzed these artifacts in both iterations to track the decision-making process of different groups and compare between the iterations.

We employed a mixed-method approach to analyze the data, combining quantitative measurement and qualitative procedures. Specifically, we used quantitative methods to measure the students' input in the evidence basket and to track changes in their decisions

throughout the project. To analyze the changes in the quality of decisions in iteration 2, we used qualitative content analysis and Wilcoxon's signed rank test (due to small sample sizes). We examined three aspects: (1) the number of pieces of evidence and their distribution of their sources and, for iteration 2 only, the contributions of evidence to the community evidence basket; (2) the quality of the initial and final decisions made; and (3) the number of changes in decisions during the learning process.

1. Evidence in the evidence basket—number and distribution of sources

To analyze the data, we counted the number of pieces of evidence that each team contributed to the evidence basket, examining their sources. If the evidence included a reference to a source of information (such as a link) without describing the evidence itself, we considered it irrelevant and did not count it. From this, we determined the total number of pieces of evidence contributed by all teams and the average number of pieces of evidence per team. We also counted the number of times different sources of evidence were cited by all teams and calculated their distribution. In iteration 2, we also performed this analysis for the community (public) evidence basket, which was a part of the collective knowledge base.

2. Quality of decisions

An evidence-based decision typically includes a claim about the proposed solution to a given issue, along with supporting evidence and a logical justification linking the two. To evaluate the quality of students' decisions, we used a scoring rubric based on the framework proposed by McNeill and Krajcik [61] (see Table 3).

**Table 3.** Scoring rubric for students' decisions.

|  | 0 | 1 | 2 |
|---|---|---|---|
| Evidence | No evidence/evidence is not scientifically based/not relevant to the decision | 1–2 scientifically based, relevant pieces of evidence | 3 or more scientifically based, relevant pieces of evidence |
| Reasoning | No reasoning | Provides reasoning, but it does not logically link evidence to decision. | Provides reasoning, that logically link evidence to decision. |

3. Changes in decisions

Throughout the project, students' teams were asked to make a decision four times in both iterations. To assess their willingness to change their decision based on further learning, we examined the changes made by each team. To ensure the reliability of the analysis, we employed a peer debriefing process that included the following steps: (a) developing a draft of the rubric, (b) conducting a joint analysis of the decisions using the draft rubric with a group of five associate researchers and discussing any disagreements, (c) refining the rubric, (d) having the five associate researchers independently analyze the decisions using the updated rubric and achieving 80% agreement, (e) analyzing the decisions with an associate researcher and achieving 90% agreement, and (f) holding a discussion until 100% agreement was reached.

## 5. Findings

### 5.1. The Characteristics of Socio-Scientific Reasoning Developed by Students in a KCI-SSI Design (RQ1)

Our analysis of the interviews with students revealed that all of them expressed themes that can be classified under at least two dimensions of socio-scientific reasoning. All students also reported that they changed the way they approach socio-scientific issues (SSIs) following the intervention. The *complexity* dimension of socio-scientific reasoning (SSR) was the most prominent, with all students expressing a variety of themes. The *multiple perspectives* and *ongoing inquiry* dimensions were less dominant but were still expressed

by most students. The *skepticism* dimension was the least demonstrated. Most students showed the ability to transfer their understanding of at least one dimension of SSR to new contexts. However, what was particularly noteworthy was the wide range of themes expressed by students, which did not fit into any of the known dimensions of socio-scientific reasoning. We therefore identified a new dimension of socio-scientific reasoning, which we label as *understanding processes of decision-making on socio-scientific issues in a community*. Table 4 presents a summary of the findings on the different dimensions of socio-scientific reasoning, including the new dimension, along with the corresponding themes. Below, we provide further details on students' expressions of socio-scientific reasoning in their own words.

**Table 4.** Classification of themes raised by interviewees in iteration 2.

| SSR Dimension(Number of Interviewees) | Theme |
|---|---|
| (1)  Recognizing the inherent *complexity* of SSIs<br><br>**Six of six interviewees** | The issue is hard to resolve<br>The issue is influenced by multiple factors<br>There is more than one solution to the issue<br>There are pros and cons to each solution<br>The issue has a wide range of influence |
| (2)  Appreciating the need for SSIs' *ongoing inquiry*<br><br>**Four of six interviewees** | The scientific knowledge about the issue is tentative<br>There is a need to deepen the knowledge about the issue<br>Investigation of the issue is needed to make a decision |
| (3)  Analyzing SSIs from multiple perspectives<br><br>**Five of six interviewees** | Different stakeholders have different perspectives on the issue<br>The issue is controversial |
| (4)  Employing *skepticism* of potentially biased information<br><br>**Two of six interviewees** | There is need to obtain information from experts |
| **(5)  The fifth dimension—understanding processes of** *decision-making on SSIs in a community*<br><br>**Six of six interviewees** | **Cooperating with others**<br>**Expressing own opinions**<br>**Sensitivity to others' opinions**<br>**Compromising to reach agreement**<br>**Willingness to consider changing decisions** |

### 5.1.1. SSR1: Recognizing the Inherent Complexity of SSIs

We identified extensive expressions of a variety of themes belonging to the *complexity* dimension in socio-scientific reasoning by *all* interviewees.

For example, Gila presented knowledge of the many factors that influence asthma, noting that following the intervention and her understanding of the multiple factors involved in the SSI, she became more aware of additional perspectives that she believes are essential for decision-making:

> *We didn't really learn until now [in the Asthma project] about how to measure traffic speed and consider the allergenic trees and the pollution from factories in the area and wind directions and all these sort of things . . . so we didn't learn about it until now so it gives another point of view and more criteria that need to be looked at . . . before making any decision.*

Gila gave an example of a similar issue, a power plant and its effect on the health of residents in the nearby neighborhood, and explained that following the project, she would deal with this issue in a different way because she understood that solutions have various advantages and disadvantages. By this, she illustrates the transfer of her understanding of the *complexity* dimension to a new context:

> *I think that before we started learning [in the Asthma project], I would have simply suggested that they move the chimneys [of the power station near the neighborhood] to another place or look for another way to generate electricity, but today when I look at it*

*with the tools we learned in the project, I realize that it has many more sides . . . . Let's say if you want to disable the work of the chimneys . . . on the one hand it can prevent a lot of cancer but on the other hand there is really no other way that is as efficient as the chimneys to generate electricity for an entire country.*

### 5.1.2. SSR2: Appreciating the Need for Ongoing Inquiry of SSIs

We found evidence of the *ongoing inquiry* dimension in the responses of four out of six interviewees. All of them attributed their insights in this dimension to the intervention in iteration 2, suggesting that the KCI-SSI approach has the potential to foster students' appreciation of the need for ongoing inquiry in SSIs.

For example, Nira expressed ideas and perhaps even transfer related to the *ongoing inquiry* dimension when she described her understanding of the need to investigate and obtain more information before making a decision about another SSI—global warming:

*I will first investigate the data I have. [I will check] if it is from a global perspective, or from a national perspective. [I will check] which place is the hottest, and which is the most polluting, and things like that. I will check my data first, which I probably wouldn't have done before we began learning [in the Asthma project].*

Gila expressed her understanding of the need to deepen the knowledge about the issue by claiming that following the intervention she will "think beyond" and "constantly look for what you don't necessarily see because it can help. Not necessarily in school or work, it can help in all kinds of things in life". It seems that she also exhibited transfer of the *ongoing inquiry* dimension when describing the uncertainty involved in the issue of the power plant: "It could be that the chimneys don't even cause it [harm to the residents' health]. It could be something else in the environment".

### 5.1.3. SSR3: Analyzing SSI from Multiple Perspectives

Five of the six interviewees expressed one or two of the themes classified under the *multiple perspectives* dimension: the perspectives of various stakeholders and the controversial nature of the issues. Two of the students specifically referred to the contribution of their learning in iteration 2 to their understanding of the different perspectives of socio-scientific issues.

Nira, for example, expressed a highly developed understanding of the *multiple perspectives* dimension when referring to the addition of fluoride to drinking water, an issue she was asked about. She described various stakeholders and the controversy involved: those who have problems with their teeth, those who want to add fluoride to the water (the health system), and those who oppose the addition of fluoride. She described in detail her own point of view:

*The problem is that many of the residents have problems with their teeth . . . and want to add fluoride to their water in general. But some oppose, because in the beginning it was dangerous, in terms of other diseases that could develop. And they also think that it slightly reduces their freedom of choice . . . as soon as fluoride is added to the water they cannot choose.*

Gila presented transfer of the *multiple perspectives* dimension when she described the various perspectives of stakeholders regarding a power plant and its effect on the health of the residents in the nearby neighborhood and specifically referred to the change in her thinking following the intervention (iteration 2).

### 5.1.4. SSR4: Employing Skepticism of Potentially Biased Information

We found only limited expression of the *skepticism* dimension. Two of the interviewees expressed the need to receive information from experts in order to make decisions regarding SSIs. These interviewees also seem to have transferred their skill to reason with the skepticism dimension in a new socio-scientific issue.

Nira, for instance, explained how she will confer with specialists in global warming to guide her decision-making: "I will then seek advice from those who are knowledgeable about these issues. that can assist me in making a wise decision".

5.1.5. SSR5: Understanding Processes of Decision-Making on SSIs in a Community

One of the interesting findings following the implementation of iteration 2 was the emergence of various themes that all interviewees mentioned when asked about their key insights from learning the unit. These themes were classified under the category of "Understanding processes of *decision-making on SSIs in a community*". Within this dimension, the two most prominent themes were students' emphasis on the need to *cooperate with others* and *compromise to reach an agreement*, which were addressed by five of the six interviewees. Another salient theme, expressed by most of the interviewees, was *sensitivity to others' opinions*. Additional themes were *willingness to consider changing decisions* and *expressing opinions*.

For instance, Gila emphasized the need to *cooperate with others* and *compromise to reach an agreement*:

> *It [the Asthma project] also promotes our ability to work in groups, deal with the opinions of others and if we end up making a group-product, and everyone has different opinions about how to bridge these gaps or find a common ground—this is something that can stay with us for our entire life.*

Rina explained how her approach to decision-making with others has evolved, moving from solely considering her own perspective to becoming more *sensitive to others' opinions*:

> *I also see the changes in my way of thinking in the beginning about how to formulate a right decision not only for myself but also for my group members, or to the people around me, and what steps I should take so that there is an effective solution, and that everyone is satisfied.*

Yasmin referred to *expressing opinions* in the group:

> *First, the whole group started discussions about it . . . we knew how to express our opinion on the matter because we also took charge of the collective map with the factories. We understood where they are, where there is an East or West wind [which affects the air pollution in the neighborhood]. Everyone explained their part and it was really fun.*

Rina referred to the willingness to consider changing decisions theme:

> *Others gave solutions and we all thought [about them]. I didn't think only about my own ideas. I suddenly thought of alternatives that I hadn't thought of before.*

*5.2. The Contribution of the KCI-SSI Approach to Students' Decision-Making Processes (RQ2)*

In this section we present our findings regarding the second research question, based on (a) the comparison between students' artifacts saved in the WISE unit in both iterations, and (b) further analysis of the six interviews with students following iteration 2. We start with a brief summary of these findings.

First, the comparison showed that those who studied the unit in iteration 2, which was designed with the KCI-SSI approach (a) developed significantly higher competencies in collecting and using evidence to support their decisions, (b) were more willing to consider changing their decisions based on new evidence, (c) relied more on their inquiry findings as a source of evidence, and (d) tended to share within the collective knowledge base the evidence they gathered, with particular emphasis on sharing the evidence they obtained through the collective inquiry processes.

Second, the analysis of the interviews strengthens these findings. When reflecting on their learning experience in the project, students prominently mentioned the design elements in the KCI-SSI and KCI approaches and especially the decision-making in mixed jigsaw groups. These experiences were mentioned by the interviewees as contributing to their learning. Furthermore, they explicitly related these design elements to what we

analyzed as their development of socio-scientific reasoning. This mainly included themes related to the fifth dimension—*decision-making on SSIs in a community*, and to the SSR1 *complexity* and SSR2 *ongoing inquiry*.

5.2.1. Analysis of Students' Evidence and Decisions in Iterations 1 and 2

The comparison between students' artifacts in the WISE platform are presented in Table 5 and discussed below.

**Table 5.** Evidence and decisions—iterations 1 and 2.

| Aspect/Iteration | Iteration 1 (18 Teams) | Iteration 2 (11 Teams) |
|---|---|---|
| Contributions to the team basket: Mean number of pieces of evidence per team (total) | 1.9 (34) | 11.3 (124) |
| Sources of evidence (distribution) Inquiry activities Authoritative source Colleagues General knowledge | 0% 59% 9% 32% | 41% 34% 10% 15% |
| Mean quality of decisions (Range = 0–2) First decision Evidence Justification | 13 decisions 0.2 0.3 | 14 decisions 0.5 * 0.6 ** |
| Final decision Evidence Justification | 0.4 0.4 | 1.7 * 1.4 ** |
| Changes in decisions (% of teams) | No change—89% 1 change—11% | No change—9% 1 change—45% 2–3 changes—46% |
| Contributions to the community basket: Mean number of pieces of evidence per team (total) | NA | 7 (77) |
| Distribution of sources of evidence Inquiry activities Authoritative source Colleagues General knowledge | NA | 72% 18% 5% 5% |

\* W = 78, *p* < 0.05 \*\* W = 33, *p* < 0.05

1.  Contributions of evidence to the team evidence basket

Despite the scaffolds throughout the unit in both iterations encouraging students to add evidence to the team basket, there was a significant difference between the two iterations. In the first, students only collected a few pieces of evidence (average of 1.9 per group) in the team evidence basket and did not rely on inquiry activities at all, instead relying mostly on external sources such as the information provided in the unit. In contrast, students in iteration 2 collected much more evidence (average of 11.3 per group) in the team basket and tended to rely on the collaborative empirical inquiry process (41%), where each group of "regional experts" collected evidence about different factors affecting asthma in their neighborhood.

2.  Quality of decisions

In both iterations, students were given support to base their decisions on evidence, such as instructions on how to use evidence to support their decisions and scaffolds to evaluate the quality of their decisions.

However, in iteration 1, we observed that the quality of students' decisions was poor, with limited reliance on evidence, and no significant improvement was observed between

the initial and final decisions (mean scoring of 0.2 for evidence and 0.3 for justification in the first decision, compared to 0.4 for both in the final decision).

In contrast, when analyzing the first and final decisions made by 14 student teams in iteration 2 using Wilcoxon's signed rank test, we observed a significant improvement in the quality of their decisions. Specifically, there was a remarkable increase in the use of evidence (mean scoring of 0.5 in the first decision and 1.7 in the final decision) and justifications (0.6 in the first decision and 1.4 in the final decision). On average, students in iteration 2 utilized 3.1 pieces of evidence to support their final decisions, all of these which emerged from the collaborative inquiry processes they conducted.

To provide an example of a high-quality decision, we present, Zamir and Mia's final decision, which we rated as having the highest score for evidence use (four pieces of evidence) and justification that logically links the evidence to the decision (Figure 4). The sources of evidence cited by these students originated in the collaborative inquiry processes, in which they collected data on factories near the city, wind directions in different areas, traffic in the morning, and the presence of allergenic plants (information regarding location of factories and frequencies of wind directions were collected via Google Maps, and information regarding traffic and allergens were collected empirically).

Based on this evidence, we recommend that the family move to Area B because it has fewer pollutants that could affect the girl's asthma. There are no factories or industrial areas in this area, the wind direction is mainly from the west, so air pollution from distant factories does not spread there, there is not much traffic (which emits polluting particles) during rush hour and in the morning, and the allergenic plants present do not include cypress (which triggers the girl's asthma).

Evidence
Justification

**Figure 4.** An example of a high-quality decision.

3.  Changes in decisions

During iteration 2, almost all student teams (91%, 10 out of 11 teams) altered their decision between 1 and 3 times (apart from one team that did not change its decision). In contrast, during iteration 1, almost all student teams (89%, 16 out of 18 teams) did not modify their decision at all, and only two teams changed their decision once. We attribute this discrepancy between the iterations to the KCI-SSI design, which fostered the cultivation of students' socio-scientific reasoning and their ability to make informed decisions grounded in the integration of evidence and multiple perspectives.

4.  Contributions of evidence to the community evidence basket

Analysis of students' contributions to the evidence basket indicated that in iteration 2, there was (a) an impressive contribution—77 pieces of evidence in total, representing more than half of the evidence that the teams collected in their team baskets (124 pieces of evidence in total) and (b) a clear preference by students to share in the community basket evidence that originated from inquiry activities (72 of 77).

5.2.2. Analysis of Students' References to Design Elements

Table 6 presents KCI and KCI-SSI design elements that were mentioned by at least three of the six interviewees following iteration 2. These design elements were mentioned as contributing to students' SSR, mainly in the SSR5 dimension.

**Table 6.** Interviewees' mentions of KCI and KCI-SSI design elements and their contribution to SSR and decision-making.

| Design Element (Number of Interviewees) | References to SSR and Decision-Making | Examples |
|---|---|---|
| Phase B of jigsaw—decision-making process in mixed groups Five of six interviewees | *Understanding processes of decision-making on SSIs in a community*: Cooperating with others Expressing own opinions Sensitivity to others' opinions Compromising to reach agreement Willingness to consider changing decisions *Analyze SSIs from multiple perspectives*: Exposure to a variety of perspectives Confidence and sense of ownership in the decision | "The activity that I liked the most . . . was the eighth work [the decision-making process in mixed jigsaw groups] . . . everyone contributed what they knew, and at the end we made a decision as a group after a discussion. It was, in my opinion, the best . . . If we would have made the decision ourselves . . . without hearing others' opinions or evidence that they collected or things they know . . . , then it would have been more difficult. But here everyone contributes and gives more information. In the end we reach a general decision that everyone agrees on, so you are almost sure of it, in what you choose". "In groups there are other people, and you can't always just say your ideas. You also have to learn from others, and you have to teach them. It must be in cooperation and compromise". |
| Using evidence from the collective knowledge base (the shared Google Map) Three of six interviewees | *Understanding processes of decision-making on SSIs in a community*: Cooperation in groups Expressing own opinions Sensitivity to others' opinions Facilitating the decision-making process | "In the group of eight, we simply looked at a collective map, opened it and saw the data of each group, then we removed the most polluted areas . . . so it was much easier that way that everyone also gave their contribution" " . . . we also took charge of the collective map with the factories. We understood where there are, where there is an East or West wind. Everyone explained their part . . . " |
| Phase A of jigsaw—collecting evidence for the collective knowledge base Three of six interviewees | Confidence in the evidence *Recognizing the inherent complexity in SSIs:* The issue is influenced by multiple factors *Appreciating the need for SSIs in ongoing inquiry:* Investigation of the issue is needed to make a decision | "For example, when we investigated the plants and all the cars, so it really showed me the large number of cars that we counted passing by in only ten minutes. This is real data . . . that I didn't even think existed . . . just seeing it with my own eyes, you know, and counting and it, it really amazed me". "it gives another point of view and more criteria that need to be taken . . . before making any decision". |

## 6. Discussion

Today's citizens often face complex, controversial, socio-scientific issues that have a great impact both on their lives and on the society in which they live [1–3]. Many of these issues are related to sustainability or to public health, such as in the recent COVID pandemic. These issues require citizens to develop opinions and make personal decisions, such as those concerning vaccination of themselves and their children or obeying social distance instructions. In our networked society, citizens' discussions regarding such issues take place more and more via the Internet, mainly in social networks [2,4]. In these venues, interactions take place with other people who may investigate, discuss, make decisions, and act upon these issues. In such contexts, as described earlier, the Internet may become a "double-edged" sword and pose new challenges in making informed decisions, while needing to deal with multiple sources of information as well as fake news [1,2,6]. In light of these challenges, it is crucial that schools will explicitly teach and deal with controversial issues and provide students with learning environments designed to foster reasoning

skills they need to make informed decisions on socio-scientific issues, while interacting with others.

The findings of this research indicate that the KCI-SSI approach that we have developed succeeded in promoting students' socio-scientific reasoning, even though researchers constantly report this as a highly difficult task [14]. Moreover, our approach revealed a fifth, undertheorized dimension of socio-scientific reasoning, which we view as especially important for citizens in the current networked society.

Specifically, the findings show that all interviewees who studied the asthma unit designed in the KCI-SSI approach (iteration 2) developed at least two dimensions of socio-scientific reasoning: they all developed the complexity dimension, and most of them also developed the multiple perspectives and ongoing inquiry dimensions. Almost all were able to transfer these reasoning skills, at least in one dimension, to new and even far contexts. Based on the literature on transfer [24,25], we view this as indicating deep understanding of the inherent characteristics of socio-scientific issues, which is necessary for the development of substantial socio-scientific reasoning. The addition of the fifth dimension, expressed in a variety of themes, related to decision-making processes regarding SSIs *as part of a community* (in this case, the classroom), also strengthens this claim.

In addition to the development of socio-scientific reasoning, including the fifth dimension, our findings indicate that students who studied via the KCI-SSI approach developed significantly better decision-making *practices* than students studying in the SSI approach. These students collected more evidence during their inquiry, tended to collect and share evidence originating from their collaborative inquiry processes, were more open-minded and willing to change their decisions, and overall made better evidence-based decisions. These findings, as we explain below, illustrate that this development was probably afforded by the KCI-SSI design.

### 6.1. The Importance of the Fifth Dimension in Socio-Scientific Reasoning

We propose five practices that encompass students' comprehension of the significance of various components in decision-making processes concerning SSIs. Here, we will elaborate on these components and their importance in decision-making within a networked society:

1. Cooperating with others. As previously discussed, cooperating with others can support argumentation [17,38] and facilitate complex decision-making processes [38]. We believe that understanding the value of collaboration is crucial for students who will engage in decision-making processes in a community.
2. Expressing own opinions. Engaging in discussions about controversial SSIs requires individuals to express and justify their opinions, defend them, and actively contribute to accountable talk within a community [53]. An attitude in which students acknowledge and appreciate listening to all voices around the table (even if they disagree with them) is a necessary condition for discussions about controversial SSIs.
3. Consideration of others' opinions. Grasping the complexities of controversial SSIs necessitates an open-mindedness towards perspectives different from one's own and the development of epistemic fluency [45]. As mentioned earlier, students often struggle with coping with others' opinions [43,44]. Therefore, understanding the importance of considering diverse viewpoints is vital in decision-making processes regarding SSIs within a community.
4. Willingness to compromise. The ability to be open to negotiation and to compromise is an integral aspect of epistemic fluency [45] and the cultivation of "reasonable disagreements" [48] among students. This research indicates that it is critical for learning in SSIs-based instruction.
5. Willingness to consider changing of decisions. The capacity to revise one's decisions, whether through new information discovered during ongoing inquiry or by being influenced by well-supported viewpoints of others, plays a critical role in decision-making concerning controversial SSIs.

We believe that students' development of the fifth dimension of socio-scientific reasoning—decision-making processes in a community—is crucial. These future citizens will, perhaps more than ever, need the capability to make decisions with others as part of online communities and collaboratively investigate socio-scientific issues while contributing to common knowledge bases that serve their communities. Given that decision-making processes in a community involve epistemic thinking about the nature of decision-making in a community, it is possible that the development of this dimension itself contributed to the improvement of students' decision-making practices within their community, as observed in the results.

It is important to note that we do not view the fifth dimension as a standalone reasoning skill. Rather, it is complementary to being aware of (a) the complexity of these issues, (b) the need for their ongoing inquiry, (c) the different perspectives they entail, and (d) the need to exercise skepticism towards information presented about them, which may be partial or biased (the original four dimensions of socio-scientific reasoning). On top of those, however, we view students' understanding of the importance of communal decision making—as delineated in the fifth dimension—as a crucial addition to the SSR literature. This is because it represents current conceptualizations of scientific literacy developed within the past decade, which emphasize engagement in public discussions on science-related issues and critical consumption of scientific information [23,26,27] which also inherently requires these skills to be transferred to new contexts [24,25].

*6.2. Supporting SSR Development and Informed Decision-Making in a Community of Learners*

Our findings highlight the potential of designing learning environments in the KCI-SSI approach to promote students' socio-scientific reasoning and decision-making in a community through a variety of processes.

1.  Collaborative inquiry and decision-making in a knowledge-building community may foster the development of SSR, including the fifth dimension. The findings from the interviews indicate prominent elements in the design of the unit that contributed to the development of socio-scientific reasoning: some of the students explicitly addressed the impact of collaborative inquiry and decision-making activities in the community on the development of their reasoning about socio-scientific issues in relation to three of the known dimensions of SSR. These findings are consistent with recommendations in the literature to support students in examining an issue from various perspectives and considering the nature of science and scientific inquiry through active involvement in in collaborative scientific inquiry [41,43,44]. Additionally, the extensive mentions that were classified as belonging to the *fifth-dimension* theme, combined with explicit mentions of KCI-SSI design elements that promoted them, further highlight the potential of the KCI-SSI approach to foster socio-scientific reasoning skills. This was especially true for the collaborative inquiry and joint decision-making in both phases of the jigsaw using the collective Google Map.

2.  Collaborative inquiry and decision-making within a knowledge-building community facilitate the development of argumentation and decision-making skills among students. The inclination of students in iteration 2 to use evidence obtained through active involvement in inquiry processes and to share it in the public evidence basket indicates a high level of confidence in this evidence. This confidence could have resulted from the fact that students played an active role in the inquiry process, including the physical collection of evidence, and is consistent with the literature regarding the importance of such active involvement in inquiry [40]. In addition, students' confidence in using this evidence, their sense of ownership, and willingness to share it with the community reflect their responsibility for the collective knowledge in the class and the development of an important aspect of accountable talk [53]. Although the students in iteration 1 had the opportunity to create collaborative arguments in teams, the quality of their arguments and decisions was unsatisfactory. These findings are supported by Michaels and his colleagues [53], who indicated that collaborative argumentation in

socio-scientific contexts does not happen automatically, and, to support it, students must develop a sense of ownership and engage in specific collaborative argumentation aspects, such as asking clarifying questions and negotiating common consent. Iteration 2, with the KCI-SSI approach, provided support such aspects through the collaborative inquiry and decision-making scripts. Additionally, design elements helped students interact with the collective knowledge base to support their decision with inquiry-derived evidence. Therefore, it appears that the KCI-SSI design promoted students' informed decision-making, supported by appropriate evidence and justification. We suggest that the design elements of the KCI-SSI approach facilitated the creation of a learning community culture in which all students are involved in a common effort to understand [52], using accountable talk [53].

3.  Collaborative decision-making in a knowledge-building community may support students' epistemic fluency. The students' willingness to change their decisions in iteration 2 indicates the development of epistemic fluency [45], which may have resulted from their advanced epistemic concepts and socio-scientific reasoning skills. According to the interviews, the KCI-SSI design helped students understand the importance of decision-making processes in the community, including the significance of being *sensitive to others' opinions* as well as being *flexible* and *willing to change decisions*. These findings suggest a relationship between the development of the fifth dimension of socio-scientific reasoning among students and the development of their epistemic fluency.

### 6.3. Implications for the Design of SSI Environments

Finally, when considering the design of SSI environments, we recommend incorporating a **collaborative inquiry script** into the SSI design to foster the development of a knowledge building community within the classroom. This script should involve students in contributing to a **collective knowledge base** using inquiry artifacts they encounter or create. This knowledge base should serve further inquiry and decision-making. Tools such as a community evidence basket or collective map can be used for this, along with other collaborative knowledge-building tools. We also suggest including **collaborative decision-making processes** in the SSI design, where the whole class functions as a community and where students express diverse opinions based on distributed expertise they gained in the project. It is crucial as part of such collaborative decision-making processes to prompt students to support their claims with robust evidence obtained from the collective knowledge base.

### 7. Limitations of the Research

This study was conducted in a private school that prioritizes project-based learning and interdisciplinary learning, emphasizing the relationship between individuals, the environment, and society. While the participating students represented a wide range of academic levels, learning disabilities, and engagement levels, the study was carried out under relatively favorable operating conditions compared to most schools. Therefore, the research results may not be generalizable to other settings and require further investigation in less favorable contexts.

We primarily utilized in-depth, semi-structured interviews with a selection of students to assess their socio-scientific reasoning. However, to increase the validity of our findings and support our conclusion regarding the development of socio-scientific reasoning in both study iterations, it would have been preferable to conduct interviews in iteration 1 as well. Furthermore, an additional quantitative tool would have allowed us to examine the students' socio-scientific reasoning before and after the intervention to better support our claims.

**Author Contributions:** Conceptualization, H.B.-H.; methodology, Y.K.; formal analysis, H.B.-H.; investigation, H.B.-H.; writing—original draft, H.B.-H.; writing—review & editing, Y.K. and T.T.; supervision, Y.K. and T.T. All authors have read and agreed to the published version of the manuscript.

**Funding:** This research received no external funding.

**Institutional Review Board Statement:** The study was conducted in accordance with the Declaration of Helsinki and approved by the Ethics Committee of the University of Haifa (protocol code 180/16, 22 May 2016).

**Informed Consent Statement:** Informed consent was obtained from all subjects involved in the study.

**Data Availability Statement:** The data presented in this study are available on request from the first author. The data are not publicly available as they represent the authors' own field material.

**Acknowledgments:** This article is based on the PhD thesis "Decision-Making on Socio-Scientific Issues in a Knowledge Community of Inquiry" submitted by Hava ben Horin in February 2020 at the Department of Learning, Instruction and Teacher Education, the Faculty of Education, University of Haifa, under the direction of Yael Kali and Tali Tal.

**Conflicts of Interest:** The authors declare no conflict of interest.

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
