# Peer review of "The Fifth Dimension in Socio-Scientific Reasoning: Promoting Decision-Making about Socio-Scientific Issues in a Community"

_sustainability, doi:10.3390/su15129708_

Round 1

Reviewer 1 Report

The manuscript is interesting and deals with 21st century important issue, ability to make competent decisions about complex socio-scientific issues which affecting individual lives, close family, society, or whole world. Complex and demanding research and the obtained results are presented in a quality and clear manner, and the conclusions are well supported by results of earlier research. The design of the learning unit and results are well presented in diagram and table format. There is a general problem with a lot of confusing reference numbers in the text, which often makes it unclear which references the authors are referring to. Also, most of the conclusions (supporting whole research question 2, and partly research question 1) were derived from the results obtained by interviewing only six of 39 students in the second iteration, i.e. 85, which represents the entire sample and this, surely, cannot be improved.

line 33

It is not clear whether the previous text refers to references [2-4] or [2,4]. For sure it is not [24].

line 39

Looks like it should be [1,5], not [1,4]. 

line 50 

Looks like it should be [7] and obviously it is not [57].

line 56

Looks like it should be [9] and obviously it is not [98].

line 61

Maybe it could  be [10]?

line 63

Unclear reference number again: [1131512].

line 67

Probably [15,16]?

line 73

Unclear reference number again: [4417].

line 79 

Unclear reference number again: [171718].

line 96 

Probably [19]?

line 98

Probably [20]?

line 106

Unclear reference number again: [2223].

line 129

Maybe it could  be [20] again?

line 133

Maybe it could  be [8]? 

lines 149, 152, 153, 169, 204, 331, 663, 681

Unclear reference numbers.

line 170

Probably [8]?

line 216, 318, 325

Missing reference number(s).

line 273

Is it “collective knowledge – base” or  “collaborative knowledge base”?

line 295

Why only six students?

line 341

“designing” instead of “de-signing”

line 519

“second iteration” or “ iteration 2” but no both

line 559

Based on the data from table 4 it is  not “0,5 and 0,6” but rather “0,4 and 0,4”.

line 562

Based on the data from table 4 it is  not “0,4 for evidence and justification” but rather “0,5 and 0,6”.

lines 576-589

As currently written, section seem duplicative. Please reorganize this to make it succinct. 

line 593

77 pieces of evidence is not “more than half” of 158.3, but rather “about half” of 158,3.

line 704

“argumentation” instead of “argumen-tation”

Author Response

  1. There is a general problem with a lot of confusing reference numbers in the text, which often makes it unclear which references the authors are referring to.

We apologize for this. The problem probably arose from the conversion of the word to PDF, we fixed this throughout the manuscript.

  1. Also, most of the conclusions (supporting whole research question 2, and partly research question 1) were derived from the results obtained by interviewing only six of 39 students in the second iteration, i.e. 85, which represents the entire sample and this, surely, cannot be improved

Thank you for this comment. We decided to conduct the in-depth interviews that we reported about after carrying out a pilot study during iterations 1 and 2, using the QaSSR questionnaire, an established tool for evaluating SSR. The pilot study did not yield significant differences in the students' SSR before and after the intervention. We have added an explanation in section 4.1 of the article detailing our decision to use instead in-depth interviews, which allowed us to gather rich data and gain a deeper understanding of the socio-scientific reasoning of a selected subset of students, who were chosen to represent a broad range of participants. Furthermore, we have included an explanation of the QaSSR tool in section 2.4 along with a reference in the new section 7 to the limitations of our study in measuring changes in SSR.

  1. Line 273 - Is it “collective knowledge base” or “collaborative knowledge base”?, line 519 - second iteration” or “ iteration 2” but no both

Thank you for noticing these inconsistencies. We have revised the entire manuscript to use the term "collective knowledge base" and "iteration 1 or 2" in order to ensure consistency in terminology.

  1. Line 559 - Based on the data from table 4 it is not “0,5 and 0,6” but rather “0,4 and 0,4, Line 562 - Based on the data from table 4 it is not “0,4 for evidence and justification” but rather “0,5 and 0,6”

There was indeed a mistake in the description of the data, and it was corrected (section 5.2.1)

  1. Line 593 - 77 pieces of evidence is not “more than half” of 158.3, but rather “about half” of 158,3

Thank you for this comment. We apologize that the data was not presented clearly enough. To provide further clarification, we would like to highlight that the total evidence collected by the 14 teams in iteration 2 and placed in the team evidence baskets was 124. The teams contributed a total of 77 pieces of evidence to the collaborative evidence basket, which accounts for more than half of the evidence collected. To address this issue, we have added a reference to the total number of evidences in the table, not just the average, and have included the total number of evidences in 5.2.1, under #4.

Reviewer 2 Report

The work is well written. However, to be applied in the real world, more explanations would be needed relative to the statistical tests used to support and validate the presented results. For example, in Table 4, page 15, it is not clear what W represents and what p<0.05 is, it is probably a P-value associated with a t-test. There are two questions or hypotheses formulated at the beginning of the paper, but it is not clear by which methods and statistical techniques to verify or validate these.

Author Response

more explanations would be needed relative to the statistical tests used to support and validate the presented results. For example, in Table 4, page 15, it is not clear what W represents and what p<0.05 is, it is probably a P-value associated with a t-test. There are two questions or hypotheses formulated at the beginning of the paper, but it is not clear by which methods and statistical techniques to verify or validate these.

Thank you for this comment. We have taken it into consideration and made several revisions to our methodology section. In Section 4.1, we have added an explanation about the data sources and the mixed method approach we used to analyze them. Additionally, we have revised Sections 4.1.1 and 4.1.2 to provide clarity on the research tools we used to answer each research question.

Specifically, we utilized semi-structured interviews to answer research questions 1 and 2 and data from WISE to answer question 2.

In Section 4.1.2, we have also provided an explanation of the statistical tool we used to analyze the data on the quality of the first and last decisions in the second iteration - Wilcoxon's signed rank test. We also added a reference to the Wilcoxon test in section 5.2.1 in #2.

Reviewer 3 Report

The manuscript approaches an educationally and socially important issue: decision making and citizenship. Moreover, the methodology and cocepts are robust and creative. 

Nevertheless, the manuscript is not appropriate for publication in the Sustainability. I suggest comments for authors to complement and revise. Please reflect my comments to the revision.

1. The authors need to state that SSI is the acronym of the socio-science issues in the last sentence of the first page.

2. Subchapters 2.1 and 2.2 is the important theoretical compotents of the manuscirpt but they are too short. Specifically, subchapter 2.1. should encompass diverse contexts and cases of scientific literacy and science education.

3. Chapters 3 and 4 should be integrated more concisely becaue both of them are chapters on the research design and methodology.

4. The discussion section has somewhat complex structure. However, it do not clearly express what the authors discuss based on the results of the study. 

5. There are several error messages: “Error! Reference sources not found.” What are they? The authors should 

Author Response

  1. The authors need to state that SSI is the acronym of the socio-science issues in the last sentence of the first page.

Thank you for noticing this problem, we've fixed this.

  1. Subchapters 2.1 and 2.2 is the important theoretical components of the manuscript but they are too short. Specifically, subchapter 2.1. should encompass diverse contexts and cases of scientific literacy and science education.

Thank you. We have expanded those sections according to your request.

  1. Chapters 3 and 4 should be integrated more concisely because both of them are chapters on the research design and methodology.

Thank you for this suggestion. Although these two chapters focus on methodology, we separated them due to the crucial role of designing the learning environment in our study. Nonetheless, we have revised them, and we hope that the connection between the two is clearer now.

  1. The discussion section has somewhat complex structure. However, it do not clearly express what the authors discuss based on the results of the study.

We have taken your feedback into account and have revised the discussion. We hope that the correlation between the findings and the discussion is now more explicit.

Reviewer 4 Report

The research addresses a relevant topic, as the development of socio-scientific reasoning is one of the important goals of science education nowadays. The novelty of the research is that it integrates proven principles for the design of learning environments for socio-scientific issues with the Knowledge Community of Inquiry model. It provides evidence-based methodological recommendations for improving decision-making abilities, considering the challenges of decision-making in a networked society. The study is important from both a theoretical and a practical perspective and is potential of broad interest.

The manuscript is high-quality, well-developed and logically structured. The research questions are theoretically sound and the methods used are appropriate to address them. The presentation of the results is clear. I propose a few clarifications and additions.

It would be useful to explain the essence of the design-based approach in more detail in the Methodology section.

Subsection 2.4 and subsection 5.2.1 also discuss the use of metacognitive scaffolding to support the decision-making process. Some examples would be useful to give the reader more information on their nature.

A Limitations and further studies section could refer to the fact that the students in this study were from a leading private school focusing on science and environmental studies. It would be useful to test the proposed KCI-SSI methodology in other schools. Furthermore, it could be investigated how the characteristics of the students (e.g. socio-cultural background, subject knowledge, epistemological knowledge, social skills, attitudes towards science) influence the effectiveness of this methodology.

Within subsection 3.3, there is only one additional subsection (3.3.1), so there is no justification for its numbered separation.

It is necessary to check the references in the text.

Author Response

  1. It would be useful to explain the essence of the design-based approach in more detail in the Methodology section

Thank you for bringing this to our attention. We agree that further clarification of DBR is necessary. As per your request, we have expanded the explanation of DBR in Section 4 - Methodology.

  1. Subsection 2.4 and subsection 5.2.1 also discuss the use of metacognitive scaffolding to support the decision-making process. Some examples would be useful to give the reader more information on their nature.

We have decided to completely remove any mention of metacognitive scaffolding since it is not crucial to the main topic of the article.

  1. A Limitations and further studies section could refer to the fact that the students in this study were from a leading private school focusing on science and environmental studies. It would be useful to test the proposed KCI-SSI methodology in other schools. Furthermore, it could be investigated how the characteristics of the students (e.g. socio-cultural background, subject knowledge, epistemological knowledge, social skills, attitudes towards science) influence the effectiveness of this methodology.

Thank you for this valuable feedback. We have taken it into consideration and have included a section on the limitations of our study. In this section, we address the issue of learner characteristics that you brought up.

  1. Within subsection 3.3, there is only one additional subsection (3.3.1), so there is no justification for its numbered separation.

We have restructured the layout of the sections in Chapter 3 in response to your comment. Thank you.

Round 2

Reviewer 3 Report

I appreciate the revised portions of the manuscript. Thank you for the revision. However, there are still issues to revise or complement. I wish the authors reflect my following comment.

1. The subsection 2.2 provides the meaning and characteristics of the SSI. I would like to recommend the authors provide more detailed explanation why SSI can contribute to the cultivation of citizenship or ciritial thinking.

2. I think the subsections 2.2 and 2.3 can be integrated into one subsection.

3. The order of the subsections 4.1 and 4.2 should be changed.

4. The authors should mention concrete and detailed data coding process and the titles of the quantitative and qualitative research methods.

5. I would like to recommend the authors to revise the discussion section focusing each dimension of the “fifth dimensions.”

Author Response

  1. The subsection 2.2 provides the meaning and characteristics of the SSI. I would like to recommend the authors provide more detailed explanation why SSI can contribute to the cultivation of citizenship or critical thinking.

Following your recommendation, we have enhanced a more detailed explanation regarding the contribution of the SSI approach to the development of citizenship and critical thinking in both section 1 (Introduction) and 2.2 (theoretical background).

  1. I think the subsections 2.2 and 2.3 can be integrated into one subsection.

Thank you for this comment. It was important for us to separate these topics in the theoretical background, as the SSI pedagogical approach (subsection 2.2) serves as the theoretical basis for the intervention we developed for this study (explained in section 3.1), while SSR (subsection 2.3) serves as the theoretical basis for the data analysis to derive the outcomes. We have also emphasized this distinction in the text.

  1. The order of the subsections 4.1 and 4.2 should be changed.

We apologize for the mistake in the numbering of the subsections. It has been corrected.

  1. The authors should mention concrete and detailed data coding process and the titles of the quantitative and qualitative research methods.

Section 4.2 and 4.2.1 have been revised to improve clarity, and a table has been added to present the data analysis methods in a more comprehensible manner.

  1. I would like to recommend the authors to revise the discussion section focusing each dimension of the “fifth dimensions.”

In accordance with your suggestion, we have included in section 6.1 a comprehensive explanation about each constituent of the fifth dimension, highlighting its importance in decision-making concerning SSIs within a community. Additionally, we have provided in section 2.4 a brief description of "reasonable disagreements" as mentioned in the updated text.

Round 3

Reviewer 3 Report

The manuscript has been meaningfully improved properly refering the comments. I would like to recommed the publication of the manuscript.